# Do Neighbors Have More Peaceful Students? Youth Violence Profiles among Adolescents in the Czech Republic, Hungary, Poland, and Slovakia

**DOI:** 10.3390/ijerph19137964

**Published:** 2022-06-29

**Authors:** Dóra Eszter Várnai, Marta Malinowska-Cieślik, Andrea Madarasová Gecková, Ladislav Csémy, Zsolt Horváth

**Affiliations:** 1Department of Clinical Psychology and Addictology, Institute of Psychology, ELTE Eötvös Lorand Univesity, 1064 Budapest, Hungary; horvath.zsolt@ppk.elte.hu; 2Methodololgy Department, Heim Pál National Insitute of Peadiatrics, 1089 Budapest, Hungary; 3Department of Environmental Health, Faculty of Health Sciences, Jagiellonian University, 31-008 Krakow, Poland; marta.malinowska-cieslik@uj.edu.pl; 4Faculty of Social and Economic Sciences, Institute of Applied Psychology, Comenius University in Bratislava, Mlynské Luhy 4, 821 05 Bratislava, Slovakia; andrea.geckova@upjs.sk; 5Department of Health Psychology and Methodology Research, Faculty of Medicine, P.J. Safarik University in Kosice, Tr. SNP 1, 040 01 Kosice, Slovakia; 6Department of Community & Occupational Health, University of Groningen, University Medical Center Groningen, A. Deusinglaan 1, 9713 AV Groningen, The Netherlands; 7Department of Public Mental Health, National Institute of Mental Health, 250 67 Klecany, Czech Republic; csemy@nudz.cz

**Keywords:** bullying, perpetration, victimization, cyberbullying, latent class analysis

## Abstract

(1) Background: Co-occurrence or overlaps of different forms or involvement in peer violence among adolescents have been broadly studied. The study aimed to assess adolescents’ violence profiles related to bullying, cyberbullying, and fighting in the Czech Republic, Hungary, Poland, and Slovakia. The study was to investigate the pattern of bullying, cyberbullying, and fighting involvement among adolescents in these four countries to test the stability of previously identified profiles. (2) Methods: We analyzed the data from the 2017/2018 international Health Behaviour in School-aged Children survey, which used proportionate sampling among adolescents aged 11–15 years old (*n* = 24,501). A Latent Class Analysis (LCA) was performed to determine violence profiles in each country. (3) Results: In Slovakia, three distinct latent classes were identified, primarily cyber victims, school bullies, and those involved in multiple forms, and in the Czech Republic, Hungary and Poland bully victims was the fourth class. (4) Conclusions: The findings suggest that peer violence prevention programs in adolescents should consider violence profiles and multiple involvements.

## 1. Introduction

Youth violence is a prevalent phenomenon, and it can take various forms (such as bullying, fighting, or cyberbullying) in school settings or young people’s lives. Bullying has been defined as negative physical or verbal actions that have hostile intent, cause distress to victims, are repeated over time (or there is a high likelihood to be repeated), and involve a power differential between bullies and their victim [1,2]. It can take many forms including physical, verbal, or other indirect forms. Cyberbullying can be defined as intentional behavior aimed at harming another person or persons through computers, cell phones, and other electronic devices, and is perceived as aversive by the victim [3,4]. Due to the nature of the Internet and current forms of technology, cyberbullying does not need to involve repeated behaviors. Fighting is also a common manifestation of peer violence that has a physical nature and implies a power balance between those involved [5]. Regarding physical fighting, the students involved are mainly of the same age and equal strength [6]. However, due to language and interpretation difficulties, aggression with or without power imbalance is sometimes hard to differentiate [7,8,9].

Peer violence has adverse outcomes on the health and social adjustment of all involved parties. Besides direct influences, long-term effects of bullying can span into later adolescence and adulthood [10]. Studies show various negative impacts of cyberbullying on adolescents’ mental health, like suicidal thinking, depressive symptoms, loneliness, frustration, and sadness, as well as difficulties in school performance, learning, and low school achievements [11]. Fighting is often associated with alcohol drinking and drunkenness, substance use, and other problem behaviors [6]. Peer violence does not only have individual-level consequences: a significant negative relationship was found between measures of positive school climate and the prevalence of student peer bullying across studies [12,13].

Regarding the dynamics of bullying, it is mostly considered as a group phenomenon with a complex interplay between different bullying roles [14]. Most studies about violence focus on the perpetrator, victim, or bully-victim role and classify children according to these traditional roles. However, there are other important bullying roles such as the supporter or assistant to the bully, the defender of the victim, or the bystanders that can represent different attitudes towards the given bullying situation. Moreover, the expansion and spillover of bullying in each community rather depends on the reaction of bystanders than the actual actions of the perpetrator [15]. On the other hand, even if children can have only one bullying role in a given situation, they might change roles across bullying scenarios and some of them can rather be classified along a bully-victim continuum [16]. Nevertheless, co-occurrence or overlaps between subtypes or involvement forms in bullying and violence have also been studied: e.g., the overlap between different victimization forms, between victims and perpetrators, between online and offline forms [17,18,19,20,21,22]. In our previous analysis, we investigated the violence involvement patterns in a nationally representative sample of adolescents using variables of bullying victimization and perpetration, cyberbullying victimization, and fighting [23]. As a result of a latent class analysis, no clear victim or perpetrator groups were identified but four classes of violence involvement were found: (1) school-aged children who are not affected by any form of violence, (2) primarily victims of online bullying, (3) students involved in school-based bullying and fighting (but not in cyberbullying involvement at all), and finally, (4) students involved in all forms of violence. The analysis was repeated on another dataset from a later data collection of the same survey, in which the cyber perpetration was also included [24]. As a result, the above-mentioned violence involvement classes were replicated and one more latent class emerged, namely school aggressors (with high involvement of school bullying perpetration and fighting). As we could replicate the same violence involvement groups across two data collections, it may be stated that violence patterns demonstrate timely stability throughout adolescence. The recent studies from violence classification confirm that students engage in complex overlapping patterns of bullying participant behaviors [25].

However, besides timely stability, geographical and cultural determinants of violence involvement patterns are also of particular interest. Therefore, we expanded our analysis to an international context and compared the violence involvement profiles of adolescents from four countries (Czech Republic, Poland, Slovakia, and Hungary, also referred to as ‘Visegrad Group’ or ‘Visegrad Four’). These countries share cultural, historical, and social similarities.

Based on the data of the Health Behaviour in School-aged Children (HBSC) survey in 2017/2018, the rate of being bullied (at least twice in the last couple of months prior to the data collection) is around or below the international average (10% for both genders) in the four countries in all age groups and both genders, except for the 11-year old students in Hungary, who showed higher prevalence for victimization in both genders [26]. Similarly, perpetration in these four countries is around the international HBSC average, which is around 5% for girls and 8% for boys [26].

Regarding cyberbullying data, the rate of those having been cyberbullied at least once was higher than the international HBSC average in Hungary and Poland in all age groups and both genders, whereas in Slovakia and the Czech Republic the cyber victimization rates are around or below average (that is, around 13%) [26]. The cyber perpetration data reflects the same tendency with above-average rates in Poland and Hungary and around or below average rates for Slovakia and the Czech Republic. Regarding fighting, another picture can be seen, as the Czech Republic demonstrates an above-international-average rate of school-aged children who have been involved in fighting at least twice in the last 12 months while other involved countries are around or below average.

The study aimed to assess the violence profiles related to bullying, cyberbullying, and fighting among adolescents from the Czech Republic, Hungary, Poland, and Slovakia. The study investigates the pattern of traditional bullying, cyberbullying, and fighting involvement in these countries to test the stability of previously identified violence profiles, including cyberbullying. The main point of the paper was to assess if there were more similarities or differences among adolescents from the four countries.

## 2. Materials and Methods

### 2.1. Sample and Procedure

The data were obtained from the international Health Behaviour in School-aged Children (HBSC) study conducted in 2017/2018. In this WHO-coordinated cross-sectional survey, each participating country is required to comply with an international research protocol that outlines standard sampling procedures, data coding, and processing methods. The translation procedure of the questionnaire followed an international HBSC survey protocol [27]. All questions were translated from the original English version into the national languages. Thereafter the translation was translated back into English by an independent translator and submitted to the HBSC Translation Hub. The translation was reviewed and accepted or modified according to the reviewer’s comments.

Nationally representative samples are generated for children aged 11, 13, and 15. The data were collected based on a stratified cluster sampling strategy and using a standardized and validated questionnaire according to the study protocol. Adolescents’ participation was anonymous and voluntary with the informed consent of school directors, parents, and students themselves. The research conformed to the principles embodied in the Declaration of Helsinki and appropriate ethical approval for the survey was obtained at the national level.

Participation was identified based on belonging to the Czech Republic, Poland, Slovakia, and Hungary. These four countries are in Central-Eastern Europe and share cultural, historical, and social similarities. The total study sample from these four countries was composed of 24,501 participants, from the Czech Republic (*N* = 11564, 49.9% female), Hungary (*N* = 3789, 52.8% female), Poland (*N* = 5224, 50.8% female), and Slovakia (*N* = 4785, 48.7% female). For age distribution of the national samples, see Table 1.

### 2.2. Measures

Bullying was measured by using the questions about perpetrating and one about being the victim using an item modified and adapted from the revised Olweus Bully/Victim Questionnaire [28]. Participants answered two questions to indicate whether they had experienced or perpetrated bullying in the past couple of months prior to the administration of the survey, with five response categories ranging from ‘I have not been bullied in this way in the past two months’ to ‘several times a week’.

Cyber perpetration was measured by the following question: ‘In the past couple of months how often have you taken part in cyberbullying (e.g., sent mean instant messages, email or text messages; wall postings; created a website making fun of someone; posted unflattering or inappropriate pictures online without permission or shared them with others)?’ Cyber victimization was measured by the question: ‘In the past couple of months how often have you been cyberbullied (e.g., someone sent mean instant messages, email or text messages about you; wall postings; created a website making fun of you; posted unflattering or inappropriate pictures of you online without permission or shared them with others)?’ The response categories were the same as in bullying.

Fighting was measured by the question: ‘During the past 12 months, how many times were you in a physical fight?’, with five frequency responses categories from ‘I have not participated in fighting in the past 12 months’ to ‘five times or more often’.

The above-mentioned factors were transformed into dichotomous indicator variables measuring perpetration: participant has bullied others at least once at school; victimization: participant has been bullied by other students at least once at school; fighting: participant has participated in fighting cyber victimization: participant has been bullied online at least once on at least one occasion during the past couple of months; cyber perpetration: the respondent participant has bullied others online at least once during the past couple of months.

In our analysis, we also included variables of participants’ age and gender. We also assessed students’ ‘life satisfaction’ with the one-item 10-point-scale of the Cantril ladder [29]. The variables of ‘perceived social support from friends and family’ are part of the Multidimensional Scale of Perceived Social Support [30]. There are four peer support items where respondents need to indicate their agreement on a seven-point-Likert scale with certain statements about help, support, and attention from friends. Perceived family support is very similar: statements refer to help and support from family members. Average item scores based on four items which measure the perceived support from friends and family were applied in the analysis. School engagement is measured by a single item measuring ‘students’ connectedness to school’ in terms of liking school with four answer options that were entered in the model as a dichotomous variable (does/does not like school).

### 2.3. Statistical Analysis

First, a descriptive analysis was carried out to present the prevalence rates and comparisons by countries in terms of the different peer violence indicators.

Second, latent class analysis (LCA) was performed to identify subgroups of adolescents with different peer violence profile characteristics. LCA was conducted separately in the involved countries because the multiple group LCA did not result in a trustworthy solution as it was not possible to replicate the best log-likelihood values even with increasing random starts. Classes were differentiated based on dichotomous indicator variables which measured if the participants (1) bullied others at school or not, (2) were bullied at school by others or not, (3) were involved in physical fights or not, (4) cyberbullied others or not, (5) were cyberbullied by others or not. An iterative estimation process was carried out where more and more complex (less parsimonious) models were fitted to the data that are in each step the number of latent classes; these were increased by one. The best-fitting model was selected in each country by considering multiple model fit indices. A model with a given number of latent classes was considered superior compared to other models if it showed the lowest values in Akaike Information Criteria (AIC), Bayesian Information Criteria (BIC), and Sample Size Adjusted Bayesian Information Criteria (SSA-BIC). Moreover, a significant result (*p* < 0.05) of the Lo–Mendel–Rubin adjusted likelihood ratio test (LMRT) indicated that a model with a given number of latent classes provides a more optimal solution compared to the model with one less latent class. Classification accuracy was measured by the Entropy index and average latent class probabilities for the most likely latent class memberships. Characteristics of the identified latent classes were determined based on item endorsement probabilities.

Thirdly, in order to validate the latent classes, multinomial logistic regression was performed separately in each country by using the R3Step method [31]. Class memberships were predicted by gender, age, life satisfaction, support from family and friends, and liking school. Due to the possible interdependence of responses within a school class, the above-mentioned analyses were corrected by considering the clustering effect. Furthermore, weighted data was used for all analyses related to the sample from Czechia in order to ensure representativeness. Analyses were carried out by Mplus 8.0 (Muthén & Muthén, Los Angeles, CA, USA) [32] and IBM SPSS Statistics softwares (IBM, Armonk, NY, USA).

## 3. Results

### 3.1. Prevalence of Peer Violence by Countries

For descriptive purposes, we provided the percentage of participants by country in relation to peer violence indicators (bullied others, being bullied, cyberbullied others, being cyberbullied, involved in fighting). We computed a series of chi-square (χ^2^) tests to examine whether there were differences by countries in the above-mentioned variables Table 2) and applied a Bonferroni correction to adjust for multiple tests (*p* < 0.001) (Table 3).

As expected, there were several differences in the forms of violence prevalence by country. Students from Hungary reported the highest rates of school bullying perpetration and also of school bullying victimization, while in this regard the lowest prevalence rates were shown in the Czech Republic. Compared to the students from the Czech Republic, adolescents in the other three countries were involved in school bullying, either as a perpetrator or victim, at significantly higher rates. In terms of physical fighting, students from the Czech Republic were involved in the highest proportion, while it was the least prevalent in Slovakia. In the Czech Republic, Hungary, and Poland prevalence rates were significantly higher concerning physical fights with others compared to Slovakia. Adolescents from Poland reported the highest rates of cyberbullying perpetration and victimization, while in this regard the lowest prevalence rates were seen in the Czech Republic. For students in Hungary, Poland, and Slovakia, cyberbullying perpetration rates were significantly higher compared to adolescents from the Czech Republic. Cyber victimization was significantly more prevalent in Hungary and Poland compared to the Czech Republic.

### 3.2. Latent Class Analysis

#### 3.2.1. Model Selection

Model fit indices for the tested models in each country are summarized in Table 4.

Czech Republic: Based on the model fit indices, a five-class model provided the lowest values in terms of AIC, BIC, and SSA-BIC, and it also showed a significantly better fit to the data than the four-class solution as per the LMRT. Therefore, a model with five latent classes is considered as the best-fitting model and was retrieved for further analyses. The average latent class probabilities for the most likely latent class memberships were 0.96, 0.70, 0.81, 0.78, and 0.94, respectively. Profile characteristics of the identified latent classes are shown in Figure 1 (CZ).

Hungary: A model with five latent classes was considered the most optimal solution as it presented the lowest rates of AIC and SSA-BIC, and the LMRT also suggested that this model offers significantly better classification than the model with four classes. The average latent class probabilities for the most likely latent class memberships were 0.93, 0.85, 0.86, 0.76, and 0.93, respectively. Profile characteristics of the identified latent classes are shown in Figure 2 (HU).

Poland: Different fit measures were not in accordance with each other in selecting the best-fitting solution. According to the BIC index, a model with four latent classes presented the most optimal fit. However, as per AIC and SSA-BIC, the five-class solution was considered the best-fitting model, in addition to the significant result of the LMRT related to this model, which indicated that a more satisfactory classification was demonstrated compared to the four-class solution. The model with six latent classes showed a weaker fit to the data based on the BIC and SSA-BIC compared to the five-class solution; however, the LMRT suggested a more optimal fit for the six-class model compared with the five-class model. The more parsimonious, five-class solution was retrieved for further analyses because a model with six latent classes did not clearly present an improvement in model fit according to most of the fit indices. The average latent class probabilities for the most likely latent class memberships were 0.82, 0.83, 0.79, 0.87, and 0.79, respectively. Profile characteristics of the identified latent classes are shown in Figure 3 (PL).

Slovakia: Considering findings of the BIC, SSA-BIC, and LMRT, a model with four latent classes was selected as the most optimal solution, compared to other solutions with three or five classes. The average latent class probabilities for the most likely latent class memberships were 0.92, 0.81, 0.75, and 0.82, respectively. Profile characteristics of the identified latent classes are shown in Figure 4 (SK).

#### 3.2.2. Description of the Classes

Czech Republic: The largest proportion of the students were assigned to Class 1 (‘Not involved in bullying’, *N* = 8674; 75.61%) which was characterized by very low probability of being involved in any form of school or cyberbullying; however, participants in this subgroup showed higher (though still mild) probability of participating in physical fights. Adolescents in Class 2 (‘Victims of school bullying’, *N* = 1484; 12.93%) reported being bullied at school with extremely high probability and they were involved in physical fights and were cyberbullied by others with moderately high and mild probabilities, respectively. Class 3 (‘Primarily victims of cyberbullying’; *N* = 116; 1.01%) incorporated individuals who (on average) showed an extremely high probability of being cyberbullied by others; this class also demonstrated moderately high and high probability of being perpetrators of cyberbullying and being involved in physical fights. Students within Class 4 (‘Perpetrators of school bullying’, *N* = 850; 7.41%) bullied others at school and were involved in physical fights with very high probability; however, they also had a mild probability of being bullied at school. Finally, Class 5 (‘Involved in multiple forms of bullying’, *N* = 349; 3.04%) illustrated a subgroup of adolescents who presented very high probability of being involved in school and cyberbullying as a perpetrator and victim as well as having physical fights (Figure 1 (CZ)).

Hungary: Class 1 (‘Not involved in bullying’, *N* = 2078; 55.12%) contained the largest number of adolescents who showed very low probability of bullying others at school and online and being victims of cyberbullying and they also reported being bullied by others at school or being involved in physical fights with low probability. Participants within Class 2 (‘Primarily victims of cyberbullying’, *N* = 339; 8.99%) were characterized by a very high probability of being cyberbullied by others and a moderate probability of being bullied at school. Students assigned to Class 3 (‘Primarily perpetrators of school bullying’; *N* = 1057; 28.04%) bullied others at school and were involved in physical fights with very high probability, but on average there was also a moderately high probability of them being bullied by others at school. Class 4 (‘Perpetrators of school and cyberbullying’; 181; 4.80%) comprised a group of adolescents who had very high probability of bullying others at school and online and participating in physical fights, though they were also cyberbullied by others with moderate probability. Individuals in Class 5 (‘Involved in multiple forms of bullying’; *N* = 115; 3.05%) were involved in school and cyberbullying either as a perpetrator or victim and physical fights as well with very high probability (Figure 2 (HU)).

Poland: Members of Class 1 (‘Not involved in bullying’; *N* = 2981; 57.13%) demonstrated very low probability of being involved in school and cyberbullying either as a perpetrator or victim and had a low probability of being involved in physical fighting. Students assigned to Class 2 (‘Victims of cyberbullying’; *N* = 526; 10.08%) were cyberbullied by others with a very high probability and showed mild probability of being bullied by others at school and of cyberbullying others. Class 3 (‘Primarily perpetrators of school bullying’; *N* = 996; 19.09%) described a subgroup of adolescents who showed moderately high probability of bullying others at school and of being involved in physical fights, while these individuals were also bullied by others at school with mild probability. Students in Class 4 (‘Primarily perpetrators of school and cyberbullying’; *N* = 440; 8.43%) were characterized by very high probability of bullying others at school and online and participating in physical fights and these students also presented moderately high probability of being victims of cyberbullying. Finally, Class 5 (‘Involved in multiple forms of bullying’; *N* = 275; 5.27%) included adolescents who showed very high probability of being perpetrators and victims of school and cyberbullying and participating in physical fights (Figure 3 (PL)).

Slovakia: Adolescents within Class 1 (‘Not involved in bullying; *N* = 2828; 65.97%) were involved in any forms of bullying with very low probability and took part in physical fights with low probability. Members of Class 2 (‘Primarily victims of cyberbullying’; *N* = 249; 5.81%) had an extremely high probability of being cyberbullied by others and a moderately high probability of being bullied by others at school, while they had low-mild probability of being perpetrators of school and cyberbullying and of being involved in physical fights. Class 3 (‘Primarily perpetrators of school bullying’; *N* = 981; 22.88%) was characterized by very high probability of bullying others at school, a moderately high probability of being involved in physical fights, and mild probability of being victims of school bullying. Finally, Class 4 (‘Involved in multiple forms of bullying’; *N* = 229; 5.34%) comprised adolescents who showed very high probability of being involved as perpetrators and victims in school and cyberbullying and taking part in physical fights (Figure 4 (SK)).

#### 3.2.3. Summary of the Latent Class Analysis

Five latent classes were recurrently identified across countries. Most of the participants (55–76%) in all countries are classified as ‘Not involved in bullying’. There is little deviation from that observable in the Czech Republic, where the non-involved group is involved in fighting with a low-moderate probability.

In all four countries, there was a group highly involved predominantly as a victim of cyberbullying. However, in Slovakia or the Czech Republic it covered only around 1% of students, whereas in Poland or Hungary the value is 9% and 10%, respectively. Nevertheless, it is evident that this group not only covers cyber victims but also victims of school bullying. The clear differentiation of online victims and school victims was only observable in the Czech Republic, but even there there was some overlap in the predominantly cyber victim group. Similarly, in the Czech Republic there was a high-moderate/high probability that this cyber victim group would become involved in fighting and cyber perpetration as well, which was not the case in the three other countries.

In all four countries, there was a group identified that was involved with a high probability as a perpetrator of school bullying and fighting. However, this group covered only 7% of students in the Czech Republic, but had a higher prevalence (19–28%) in the other three countries. This profile was mainly similar across nations; nevertheless, in Hungary, this group was also involved with a moderate-high probability of being a victim of school bullying; meanwhile, elsewhere it was only a low-moderate likelihood for this group membership.

In Poland and Hungary there was a group where adolescents were involved mainly as aggressors (school and cyber perpetrators with the highest likelihood of participating in fighting or as cyber victims with moderate likelihood). However, the prevalence here is a little bit higher in Poland (8%) than in Hungary.

Finally, in all countries, the group of students involved in multiple forms of violence is clearly observable (3–5%).

### 3.3. Validation of the Latent Classes

Table 5 presents the results of the multinomial logistic regression analyses in each country. The ‘Not involved in bullying’ class was selected as a reference category in each country-specific analysis.

#### 3.3.1. The Czech Republic

Compared to the ‘Not involved in bullying’ class, the female gender, between ages of 10.5 and 12.5 years (compared to ages between 14.6 and 16.5 years), with lower levels of life satisfaction and support from friends, and not liking school predicted higher odds for membership of the ‘Victims of school bullying’ class. Being male and having lower levels of life satisfaction and support from friends were associated with higher odds of being assigned to the ‘Primarily victims of cyberbullying’ class, compared to the reference category. In the case of the ‘Perpetrators of school bullying’ subgroup, as opposed to the reference class, higher odds for membership were associated with male gender, age between 10.5 and 12.5 years (compared to age between 14.6 and 16.5 years), lower levels of life satisfaction and support from friends and not liking school. Finally, class membership of ‘Involved in multiple forms of bullying’ was predicted by male gender, age between 10.5 and 12.5 years (compared to age between 14.6 and 16.5 years), lower levels of life satisfaction and support from family, and not liking school.

#### 3.3.2. Hungary

Female gender, lower levels of life satisfaction and support from family, and not liking school predicted higher odds to be assigned to the ‘Primarily victims of cyberbullying’ class, compared to the reference class. Higher odds for membership of the ‘Primarily perpetrators of school bullying’ was associated with male gender, age between 10.5 and 12.5 years (compared to both age groups: between 12.6–14.5 and 14.6–16.5 years), lower level of life satisfaction, and not liking school. Compared to the ‘Not involved in bullying’ class, male gender, age between 14.6 and 16.5 years (compared to age between 10.5 and 12.5 years), lower level of support from family, and not liking school predicted higher odds of being a member of the ‘Perpetrators of school and cyberbullying’ class. Members of the ‘Involved in multiple forms of bullying’ subgroup were more likely to be males, between the age of 10.5 and 12.5 years (compared to both age groups: between 12.6–14.5 and 14.6–16.5 years), who showed lower rates of life satisfaction and support from family, and did not like school, as opposed to the reference category.

#### 3.3.3. Poland

Lower levels of life satisfaction and support from family and friends predicted higher odds for membership in the ‘Victims of cyberbullying’ class, compared to the reference class. In the case of the ‘Primarily perpetrators of school bullying’ subgroup, the male gender, between the ages of 10.5 and 12.5 years (compared to ages between 14.6 and 16.5 years), with lower rates of life satisfaction and support from family and friends were positively associated with the class membership, compared to the ‘Not involved in bullying’ class. Higher odds for membership of the ‘Primarily perpetrators of school and cyberbullying’ was associated with male gender, age between 12.6 and 14.5 years (compared to age between 10.5 and 12.5 years), lower level of support from family, and not liking school. Compared to the reference category, male gender, age between 10.5 and 12.5 years (compared to age between 14.6 and 16.5 years), lower levels of life satisfaction and support from family and friends, and not liking school were associated with higher odds of assignment to the ‘Involved in multiple forms of bullying’ class.

#### 3.3.4. Slovakia

Adolescents within the ‘Primarily victims of cyberbullying’ class were more likely to be females, between the age of 10.5 and 12.5 years (compared to ages between 14.6 and 16.5 years), with lower rates of life satisfaction and support from family and friends, compared to the reference class. Male students who had lower levels of life satisfaction and support from family and did not like school had higher odds of membership in the ‘Primarily perpetrators of school bullying’ subgroup. In the case of the ‘Involved in multiple forms of bullying’ class, membership was positively associated with male gender, lower levels of support from family and friends, and not liking school, compared to the ‘Not involved in bullying’ class.

#### 3.3.5. Summary of Validation Analysis

Even in the validation analysis, similarities were found between countries (that emerged at least in two countries).

In the case of the ‘involved in multiple forms of bullying’ class, being male, having lower family support, and disliking school predicted group membership. Apart from Slovakia, this group membership was also associated with younger age (compared to older students (14.5–16.5 years old)) and lower life satisfaction. In Slovakia and Poland, it was also significantly associated with lower peer support.

The ‘perpetrators of school and cyberbullying’ groups were similar in Poland and Hungary. Here, group membership was predicted by being male, with low family support, and disliking school. Older age was consistently associated with being an aggressor but in Poland it was significant compared to the age group of 12.5–14.5 years old, whereas in Hungary a significant association resulted when compared to the oldest age group (14.5–16.5 years old).

Members of the group ‘primarily school perpetrators’ were predicted by being male and having lower life satisfaction in all countries. Except for Slovakia, this group membership was predicted by older age (14.5–16.5). In Poland, those who dislike their school had higher odds of being a school perpetrator. Finally, the effect of family support was significant in Slovakia and Poland, whilst the effect of peer support was significant in the Czech Republic and Poland.

In the case of the group ‘primarily involved as cybervictims’, there were some ambiguities observed. On the one hand, in Slovakia and Hungary girls had higher odds of belonging to this group, whereas in the Czech Republic boys were mainly involved. However, this latter phenomenon could be explained by higher cyber perpetration prevalence. Life satisfaction was a significant predictor here for all countries. Except for the Czech Republic, low family support predicted the membership of this group significantly. Peer support was significantly associated too, apart from Hungary.

## 4. Discussion

According to our results, despite some small differences, similar latent classes were recurrently identified across the involved four countries. In particular, the violence involvement profiles of Poland and Hungary were very similar. These latent classes are ‘non-involved children’, a class of those ’involved in multiple forms of violence’, and the remaining classes reflect a versatile overlap of violence involvement forms. Uniquely in the Czech Republic, a group of school victims and separately a group of cyber victims were identified. In other countries, school victims and cyber victims together constitute a separate group. Interestingly, all of the groups in all countries are, even if only to a small extent, involved in fighting, confirming that fighting and bullying are distinct forms of violence [33]. It is also notable that in the group of ‘predominantly school perpetrators’ students are also involved in victimization. This is in line with other findings of frequent overlap between perpetration and victimization [34,35]. Surprisingly, no clear cyber perpetrator or a cyberbully–cybervictim group was identified in any country; however, there is extensive literature on the description of cyberbullying perpetrators and it is also documented that one crucial predictor of cyber perpetration is the previous online victimization [36,37,38].

The reason for similarities may be twofold. On the one hand, these latent classes might universally represent timely and geographical stability. The previous studies on overlaps between cyberbullying and school bullying or perpetration and victimization all revealed somewhat similar classes [17,25]. As was seen in the baseline data, students who are not involved in any forms of violence constituted the highest proportion of respondents. According to the latest HBSC international report, in the participating countries even despite significant geographical differences the majority of children reported that they had not been bullied regularly at school [26]. However, the ‘non-involved’ group does rather reflect non-direct involvement. First, no measure for bystander role, defender role, or bully-supporter role was included in our study, and children in these roles might also be part of the ‘non-involved class’. Secondly, it also has to be taken into consideration that the scientific interpretation of bullying (that is used in large-scale studies) is not necessarily the definition or the interpretation children use when they have to make decisions concerning their involvement [9,39,40,41]. The willingness to report about possible perpetration or victimization may depend on age, study methodology, social consciousness about bullying, and also on the presence or absence of antibullying prevention programs [42,43]. According to a meta-analysis, the antibullying interventions have significant positive effects on increasing the reporting of bullying occurrences [44].

The group of children that are ‘involved in all forms of violence’ was also found to be universal. An extensive literature describes the correlates of being a bully-victim; however, not only being a bully and a victim at once, but being involved both in offline and online violence might further increase the adjustment problems of the affected students [34,45,46]. These children may, to a probably higher extent than only offline bully victims, suffer from emotional maladjustment problems, over-arousal, impulse control difficulties, or even psychiatric disorders [45]. They also might be involved in so-called challenging bullying cases as being resistant to bullying prevention and intervention efforts [44].

The other reason for similarities of the violence involvement patterns might lie at the social-macrosystem level. Espelage proposed the use of Bronfenbrenner’s social-ecological theory to understand the characteristics of bullies and the consequences of bullying [45,47]. It posits that bullying interaction occurs not only because of individual characteristics of the child who is bullying, but also because of actions of peers, teachers, and other adult caretakers at school, physical characteristics of the school grounds, family factors, cultural characteristics, and even community factors [48]. According to the literature, together with individual, family, peer, and school-level factors, exo- and macrosystem variables such as mass media, emphasis on academic achievement, collectivism vs. individualism, antibullying school policies, school size, number of teachers may also have an impact on bullying [49,50,51]. The detailed review of similarities and differences of the involved countries along with cultural aspects, the structure and functioning of the school system, the antibullying efforts, and the societies’ general attitude towards violence stretches further than this study. However, regarding the cultural and social effects of bullying, it should be noted that none of the involved countries have a single word in their native language for bullying that can be resulted in interpretation problems [39]. Another similarity is that socioeconomic status does not show a strong association with any analyzed violence form in any of these countries [26]. Additionally, the time trends for prevalence rates regarding bullying remained relatively stable in the Czech Republic or Poland, whereas in Hungary a slight increase was observed between 2002 and 2014 [22].

As for future directions, it would be interesting to investigate further macro-level variables, especially the effect of bullying prevention approaches on the violence involvement patterns. Why students can engage in multiple forms and what is the mechanism of changing the bullying roles are also interesting topics.

### Limitations

In our analysis, only five items were available in the protocol measuring bullying, cyberbullying, and fighting. We further applied dichotomous division and classified respondents as ‘involved’, and ‘not involved’. Collection and analysis of the data regarding frequency, forms, and circumstances related to peer violence behaviors among adolescents would have been worthwhile to carry out, as this would allow estimating the dose effect, which is the exposure and severity [52].

A further limitation is that this cross-sectional study indicates relationships between variables and does not provide grounds for conclusions based on cause and effect.

Another objection might be that we used self-reported data to determine bullying status: there might be over- or underreporting of the prevalence of bullying. For example, bullies may be proud of their successful attacks and therefore overreport their bullying actions, whereas victims may be ashamed or afraid which could lead to underreporting their exposure (Hopkins, Taylor, Bowen and Wood, 2013).

It is also possible that children have an insecure understanding of peer violence. Indeed, research suggests that children fail to accurately identify cases of bullying and there are inconsistencies between the understanding held by academics compared to young people (Hopkins et al., 2013). Several studies confirmed that young participants did not consistently report the criteria of repeated behaviors, intentionality, or a power imbalance and students are more likely to mention the behaviors involved rather than the criteria by which they define these behaviors [40]

Besides the limitations, our study contributed to the understanding that involvement in peer violence has different patterns than conventional theoretical classes.

## 5. Conclusions

The current study investigated bullying, cyberbullying, and fighting behavior among adolescents in four neighboring countries: the Czech Republic, Poland, Slovakia, and Hungary. In the Czech Republic, Poland, and Hungary, five, and in Slovakia four, latent classes were identified based on respondents’ involvement in different peer violence forms. In all countries, the largest group was the ‘not involved in any forms of violence’ group. Besides this group two other groups were identified in all countries: ‘Involved in multiple forms of bullying’ group and ‘Primarily victims of cyberbullying’. One further latent class was similar in Poland, Slovakia, and Hungary, resulting in the group ‘Primarily perpetrators of school bullying’. Poland and Hungary shared one more additional group: ‘Perpetrators of school and cyberbullying’ groups. Interestingly, a clear school victim group was only identified in the Czech Republic. Our study clearly shows that most of the children—across different countries—are involved in multiple roles in bullying.

So far in prevention activities, there has been an important focus on emphasizing the victim’s suffering. Multiple involvements imply that even if students are affected as victims, it will not prevent them from acting as perpetrators. It is important that on the cognitive level of antibullying programs students should be educated about the possibility of multiple involvement. Teachers or members of school antibullying teams should be encouraged to intervene even if students change their roles across bullying situations.

Future research should continue to examine the co-occurring bullying participant behaviors and explore potential outcomes and interventions for students who are broadly involved in bullying participant behaviors.

## Figures and Tables

**Figure 1 ijerph-19-07964-f001:**
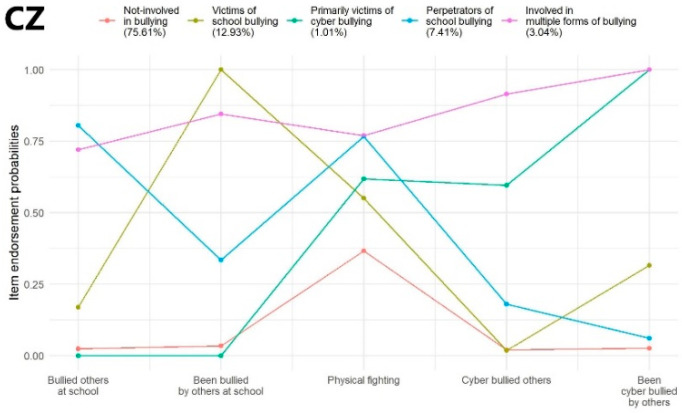
Latent class profile of Czech Republic.

**Figure 2 ijerph-19-07964-f002:**
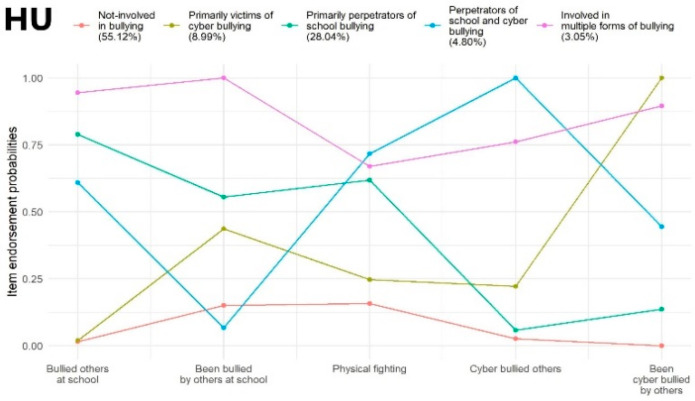
Latent class profile of Hungary.

**Figure 3 ijerph-19-07964-f003:**
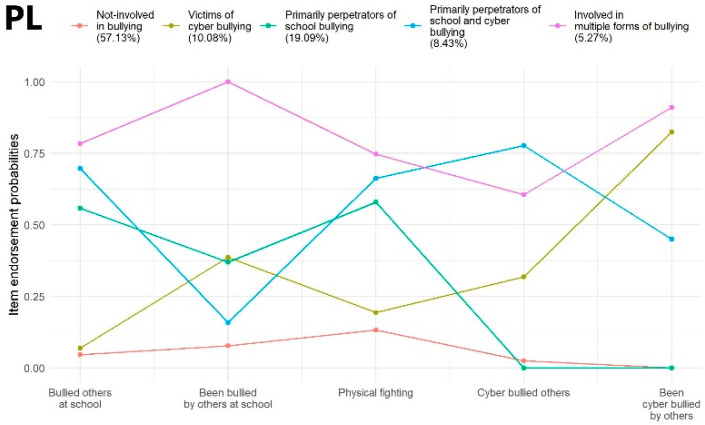
Latent class profile of Poland.

**Figure 4 ijerph-19-07964-f004:**
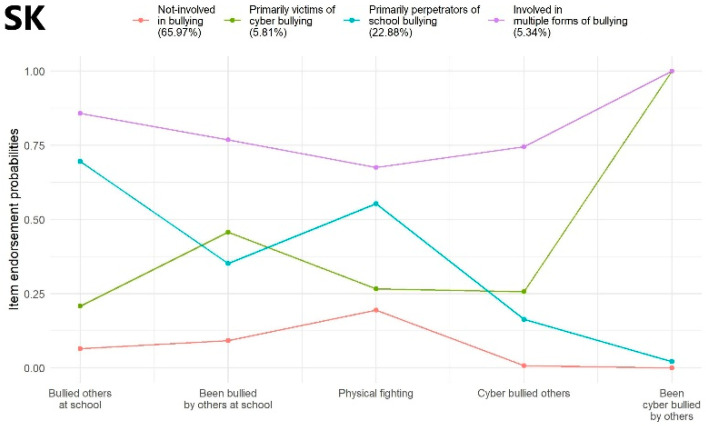
Latent class profile of Slovakia.

**Table 1 ijerph-19-07964-t001:** Age distribution of the total samples by country.

	Czech Republic	Hungary	Poland	Slovakia
Age between 10.5 and 12.5	3770 (32.6%)	1259 (33.4%)	1710 (32.8%)	1586 (33.1%)
Age between 12.6 and 14.5	3982 (34.4%)	1368 (36.8%)	1726 (33.1%)	1906 (39.8%)
Age between 14.6 and 16.5	3812 (33.0%)	1145 (30.4%)	1781 (34.1%)	1293 (27.0%)
*N* (Total)	11,564	3789	5224	4785

**Table 2 ijerph-19-07964-t002:** Prevalence rates and overall differences in the distributions of peer violence indicated by country.

	Czech Republic	Hungary	Poland	Slovakia	χ^2^ (*p*)
Bullied others at school *N* (%)	1504 (13.28%)	1158 (31.00%)	1422 (27.37%)	1088 (25.76%)	813.06 (<0.001)
Been bullied at school by others *N* (%)	2082 (18.48%)	1217 (32.65%)	1221 (23.49%)	868 (20.61%)	338.15 (<0.001)
Physical fights with others *N* (%)	4895 (43.82%)	1305 (34.82%)	1722 (33.14%)	1283 (30.51%)	329.28 (<0.001)
Cyberbullied others *N* (%)	824 (7.36%)	478 (12.82%)	845 (16.25%)	399 (9.52%)	326.16 (<0.001)
Been cyberbullied by others *N* (%)	1168 (10.44%)	676 (18.13%)	974 (18.72%)	469 (11.15%)	299.57 (<0.001)

Note. χ^2^ = Chi square statistics indicating overall differences in the distributions between the countries. Percentages in each cell show prevalence rates within the given country. Peer violence indicators are dichotomous variables, coded as 0 = No and 1 = Yes.

**Table 3 ijerph-19-07964-t003:** Comparisons of the countries in terms of peer violence indicators.

	Czech Republic	Hungary	Poland	Slovakia
Bullied others at school *N* (%)	Ref.	**2.93** **[2.69–3.20]**	**2.46** **[2.27–2.67]**	**2.27** **[2.08–2.47]**
Been bullied at school by others *N* (%)	Ref.	**2.14** **[1.97–2.32]**	**1.35** **[1.25–1.47]**	**1.15** **[1.05–1.25]**
Physical fights with others *N* (%)	**1.78** **[1.65–1.92]**	**1.22** **[1.11–1.34]**	**1.13** **[1.03–1.23]**	Ref.
Cyberbullied others *N* (%)	Ref.	**1.85** **[1.64–2.08]**	**2.44** **[2.20–2.70]**	**1.32** **[1.17–1.50]**
Been cyberbullied by others *N* (%)	Ref.	**1.90** **[1.71–2.11]**	**1.98** **[1.80–2.17]**	1.08 [0.96–1.21]

Note. OR (95% CI): Odds ratio (95% confidence interval). Ref.: Reference category. For all variables, the country with the lowest prevalence rate was specified as a reference category. Peer violence indicators are dichotomous variables, coded as 0 = Absence and 1 = Presence (the odds ratios represent the odds for the presence of the given outcome in the given country). Odds ratios and confidence intervals presented with bold figures are significant, at least *p* < 0.05.

**Table 4 ijerph-19-07964-t004:** Model fit indices for the tested models with increasing numbers of latent classes in each country.

	AIC	BIC	SSA-BIC	Entropy	LMRT	LMRT-p
**Czech Republic**						
1-class model	48,380.77	48,417.51	48,401.62	-	-	-
2-class model	44,888.79	44,969.62	44,934.66	0.74	3442.60	<0.001
3-class model	44,477.00	44,601.91	44,547.88	0.78	416.37	<0.001
4-class model	44,318.68	44,487.67	44,414.58	0.81	167.34	<0.001
**5-class model**	**44,262.29**	**44,475.37**	**44,383.21**	**0.86**	**67.19**	**<0.001**
6-class model	44,264.69	44,521.86	44,410.63	0.87	9.43	0.275
**Hungary**						
1-class model	20,574.19	20,605.37	20,589.48	-	-	-
2-class model	19,215.20	19,283.78	19,248.83	0.61	1343.80	<0.001
3-class model	18,961.62	19,067.61	19,013.59	0.82	260.31	<0.001
4-class model	18,905.71	19,049.12	18,976.03	0.84	66.56	<0.001
**5-class model**	**18,876.68**	**19,057.49**	**18,965.34**	**0.82**	**40.22**	**0.001**
6-class model	18,882.70	19,100.92	18,989.71	0.75	5.86	0.294
**Poland**						
1-class model	28,009.05	28,041.85	28,025.96	-	-	-
2-class model	26,239.72	26,311.88	26,276.93	0.59	1747.31	<0.001
3-class model	25,847.33	25,958.85	25,904.83	0.72	396.67	<0.001
4-class model	25,728.13	25,879.01	25,805.92	0.76	128.69	<0.001
**5-class model**	**25,697.21**	**25,887.45**	**25,795.30**	**0.69**	**42.10**	**0.022**
6-class model	25,697.22	25,926.81	25,815.59	0.68	11.77	0.044
7-class model	25,708.22	25,977.17	25,846.89	0.76	0.98	0.452
**Slovakia**						
1-class model	19,865.02	19,896.84	19,880.95	-	-	-
2-class model	18,324.25	18,394.24	18,359.29	0.72	1522.44	<0.001
3-class model	18,148.24	18,256.42	18,202.40	0.74	184.33	<0.001
**4-class model**	**18,109.75**	**18,256.10**	**18,183.02**	**0.77**	**49.51**	**<0.001**
5-class model	18,102.73	18,287.27	18,195.12	0.71	18.64	0.057

Note. AIC = Akaike Information Criteria; BIC = Bayesian Information Criteria; SSA-BIC = Sample size-adjusted Bayesian Information Criteria; LMRT = Lo–Mendel–Rubin adjusted likelihood ratio test. The selected model in each country is highlighted in bold font.

**Table 5 ijerph-19-07964-t005:** Multinomial logistic regression analyses in each country to examine predictors of the latent classes.

Czech Republic (*N* = 10488)	Victims of School Bullying (12.93%) OR (95% CI)	Primarily Victims of Cyberbullying (1.01%) OR (95% CI)	Perpetrators of School Bullying (7.41%) OR (95% CI)	Involved in Multiple Forms of Bullying (3.04%) OR (95% CI)
Gender ^1^	**1.514** **[1.238–1.853]**	**0.537** **[0.316–0.911]**	**0.250** **[0.187–0.332]**	**0.709** **[0.539–0.933]**
Age: between 12.6 and 14.5 years ^2^	0.931 [0.745–1.165]	1.311 [0.695–2.475]	0.995 [0.737–1.343]	0.760 [0.546–1.059]
Age: between 14.6 and 16.5 years ^2^	**0.500** **[0.383–0.651]**	1.165 [0.608–2.234]	**0.649** **[0.472–0.891]**	**0.548** **[0.380–0.789]**
Life satisfaction	**0.778** **[0.741–0.817]**	**0.807** **[0.718–0.908]**	**0.874** **[0.824–0.927]**	**0.778** **[0.714–0.848]**
Support from family	1.007 [0.995–1.019]	1.001 [0.972–1.031]	1.000 [0.986–1.014]	**0.970** **[0.955–0.986]**
Support from friends	**0.975** **[0.964–0.987]**	**0.965** **[0.931–0.999]**	**0.980** **[0.965–0.996]**	0.994 [0.977–1.012]
Liking school ^3^	**0.432** **[0.353–0.527]**	0.704 [0.412–1.202]	**0.515** **[0.401–0.660]**	**0.440** **[0.329–0.590]**
**Hungary** **(*N* = 3547)**	Primarily victims of cyberbullying (8.99%) OR [95% CI]	Primarily perpetrators of school bullying (28.04%) OR [95% CI]	Perpetrators of school and cyberbullying (4.80%) OR [95% CI]	Involved in multiple forms of bullying (3.05%) OR [95% CI]
Gender ^1^	**1.647** **[1.178–2.303]**	**0.436** **[0.351–0.542]**	**0.404** **[0.239–0.683]**	**0.583** **[0.360–0.945]**
Age: between 12.6 and 14.5 years ^2^	0.782 [0.521–1.173]	**0.517** **[0.374–0.714]**	1.887 [0.767–4.640]	**0.456** **[0.266–0.780]**
Age: between 14.6 and 16.5 years ^2^	0.890 [0.584–1.356]	**0.237** **[0.166–0.338]**	**2.570** **[1.104–5.982]**	**0.416** **[0.215–0.803]**
Life satisfaction	**0.864** **[0.794–0.940]**	**0.925** **[0.867–0.987]**	0.904 [0.756–1.080]	**0.805** **[0.716–0.905]**
Support from family	**0.963** **[0.933–0.993]**	0.972 [0.942–1.003]	**0.944** **[0.895–0.995]**	**0.916** **[0.882–0.951]**
Support from friends	0.981 [0.951–1.012]	0.979 [0.956–1.003]	1.014 [0.962–1.069]	0.979 [0.936–1.024]
Liking school ^3^	**0.658** **[0.469–0.923]**	**0.656** **[0.507–0.848]**	**0.499** **[0.291–0.857**	**0.519** **[0.321–0.840]**
**Poland** **(*N* = 5006)**	Victims of cyberbullying (10.08%) OR [95% CI]	Primarily perpetrators of school bullying (19.09%) OR [95% CI]	Primarily perpetrators of school and cyberbullying (8.43%) OR [95% CI]	Involved in multiple forms of bullying (5.27%) OR [95% CI]
Gender^1^	1.229 [0.928–1.626]	**0.300** **[0.231–0.389]**	**0.328** **[0.249–0.431]**	**0.304** **[0.210–0.440]**
Age: between 12.6 and 14.5 years ^2^	1.001 [0.732–1.370]	1.162 [0.849–1.590]	**1.853** **[1.218–2.819]**	0.685 [0.442–1.063]
Age: between 14.6 and 16.5 years ^2^	0.754 [0.554–1.026]	**0.572** **[0.408–0.801]**	1.490 [0.993–2.236]	**0.329** **[0.199–0.543]**
Life satisfaction	**0.863** **[0.806–0.925]**	**0.893** **[0.832–0.958]**	0.964 [0.884–1.050]	**0.823** **[0.743–0.911]**
Support from family	**0.965** **[0.942–0.988]**	**0.967** **[0.942–0.992]**	**0.931** **[0.910–0.954]**	**0.923** **[0.893–0.954]**
Support from friends	**0.964** **[0.943–0.985]**	**0.974** **[0.954–0.996]**	1.016 [0.994–1.038]	**0.964** **[0.932–0.996]**
Liking school ^3^	0.848 [0.633–1.135]	0.899 [0.669–1.209]	**0.693** **[0.522–0.919]**	**0.446** **[0.296–0.671]**
**Slovakia** **(*N* = 3142)**	Primarily victims of cyberbullying (5.81%) OR [95% CI]	Primarily perpetrators of school bullying (22.88%) OR [95% CI]	Involved in multiple forms of bullying (5.34%) OR [95% CI]	
Gender ^1^	**2.460** **[1.522–3.976]**	**0.559** **[0.421–0.741]**	**0.524** **[0.291–0.944]**	
Age: between 12.6 and 14.5 years ^2^	0.782 [0.511–1.196]	1.323 [0.921–1.901]	1.342 [0.718–2.507]	
Age: between 14.6 and 16.5 years ^2^	**0.487** **[0.298–0.797]**	1.250 [0.838–1.864]	0.900 [0.449–1.805]	
Life satisfaction	**0.792** **[0.713–0.881]**	**0.897** **[0.831–0.968]**	0.970 [0.817–1.153]	
Support from family	**0.963** **[0.938–0.988]**	**0.967** **[0.944–0.990]**	**0.924** **[0.896–0.953]**	
Support from friends	**0.968** **[0.941–0.994]**	0.984 [0.963–1.006]	**0.946** **[0.911–0.981]**	
Liking school ^3^	0.708 [0.459–1.092]	**0.668** **[0.501–0.889]**	**0.578** **[0.350–0.952]**	

Note. In each country, the not involved in bullying class was specified as a reference category. OR (95% CI): Odds Ratio (95% Confidence Interval). Odds Ratios and Confidence Intervals presented with bold figures are significant, at least *p* < 0.05 level. ^1^ Coded as: 1 = male, 2 = female. ^2^ Reference category: age between 10.5 and 12.5 years. ^3^ Coded as: 0 = Do not like school, 1 = Like school at least a bit.

## Data Availability

The data are not publicly available due to internal HBSC data access policy. Data access to previous HBSC rounds is provided by the HBSC Data Management Centre—Department of Health Promotion and Development, University of Bergen (https://www.uib.no/en/hbscdata).

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
