# Peer review of "Do Neighbors Have More Peaceful Students? Youth Violence Profiles among Adolescents in the Czech Republic, Hungary, Poland, and Slovakia"

_ijerph, 2022, doi:10.3390/ijerph19137964_

Round 1

Reviewer 1 Report

Nice job! Very interesting and a good contribution to understanding the phenomenon of peer violence.

I suggest considering the relevance of including in the summary information on the variables used to characterize or predict participation in each group of students.

Author Response

Dear Reviewer,

please see the attachment uploaded.

Thank you for your review. 

Reviewer 2 Report

Dear authors

Your study has relevance as a comparative perspective between countries.

Please, consider reviewing some aspects in the text edition as I show you as follows:

- Title: question mark and two-point together.

- Line 56: remove the space that is not necessary.

- Lines 136, 140, 152, 153 and 160, among others, be consistent with the quotes (double or single).

- Neither the introduction nor the conclusion refers to determinant roles in bullying, such as passive observers or victim's defenders. There is a lack of reference to the class climate as a whole environment at school which is affected by peer violence. Therefore, the limitations shown in lines 473 and 482 are far more relevant.

Please, consider including a few references to complete your vision of the topic.

Author Response

Dear Reviewer,

thank you very much for your comments.

Please see attachment uploaded.

Reviewer 3 Report

This article aimed to assess the violence profiles related to bullying, cyberbullying and fighting among adolescents of four Central-Eastern Europe countries. It is interesting to note that the data deal with both aspects of perpetration and victimization and that they are from a relatively large sample of four countries. A comparison of online victimization and interpersonal violence, which has been the focus of much attention in recent years, is also of interest. The statistical analyses also appear to have been performed well. However, this manuscript has major problems with the structure of the paper. Moreover, it is difficult to see the point on which the paper really wants to focus.

Currently, the results for each of the four countries are listed in the Results section, and a comparison of the four countries is listed in the Discussion section. The Results section is formatted with figures and tables, and the results of the figures and tables are noted in the text. The manuscript is redundant, as what is shown in the figures and tables is duplicated in the text. If the Results section presents the results for each country in figures and/or tables, the text could compare the four countries and show how they differ without interpretation.

The Discussion section is the place to make comparisons and compare important points with existing papers. The authors' interpretations of the results are also included in the Discussion section. It is recommended that the manuscript be reconstructed according to these rules.

Since readers may have a specific interest in knowing more about the characteristics in cyberbullying, one idea would be to focus on these aspects. In addition, it would be interesting to consider whether or not there is a chain of events between the victimization and the perpetration of the violence. Regarding the country-by-country comparisons, I thought it would be good to know from the Discussion whether there were more similarities or differences among the four countries, and what measures should be considered based on the results.

Finally, the four countries are marked with "V4" as a political distinction, but from a foreigner's point of view, I could not understand what it meant. If it is to emphasize Central-Eastern Europe, it should explain more that V4 is a Visegrad Group, and if it is not so relevant, it would be OK to omit this "V".

Author Response

Dear Reviewer,

thank you for your review, your comments were really very helpful. 

Please see attachement uploaded for our detailed answer.

Thank you,

the corresponding author

Round 2

Reviewer 3 Report

The manuscript has been much improved. I appreciate the authors' great effeorts that they have made in response to my questions and concerns.